# Harmonization of Amoxicillin Dose, Duration, and Formulation for Acute Childhood Respiratory Infections

**DOI:** 10.3390/antibiotics12071138

**Published:** 2023-06-30

**Authors:** Dhanya Dharmapalan, Julia Bielicki, Mike Sharland

**Affiliations:** 1Consultant in Pediatric Infectious Diseases, Apollo Hospitals, Navi Mumbai 400614, India; 2Centre for Neonatal and Paediatric Infection, Institute of Infection and Immunity, St. George’s University of London, Cranmer Terrace, London SW17 0RE, UK; jbielick@sgul.ac.uk (J.B.); msharland@sgul.ac.uk (M.S.)

**Keywords:** guidelines, respiratory infections, dispersible tablets, syrups, AWaRe book

## Abstract

Pediatric guidelines vary in their recommended amoxicillin dosing for common respiratory infections. It would help program delivery if there was harmonization of dosing and formulation of amoxicillin across multiple clinical respiratory infections, considering the pharmacokinetics, common targets, drug resistance, availability, cost effectiveness, and ease of administration. The World Health Organization EML AWaRe Book recommends higher dose amoxicillin given twice daily for five days for all uncomplicated respiratory infections where an antibiotic is indicated. The WHO AWaRe Book amoxicillin dosing guidance can be achieved for infants and older children using only scored 250 mg and 500 mg dispersible tablets (DTs), the WHO recommended child formulation. There is a clear need for wider availability of 250 mg/500 mg dispersible tablets of amoxicillin in both public and private health care sectors, to improve access to essential antibiotics.

## 1. Introduction

Childhood pneumonia remains a leading cause of mortality in children globally. Access to standardized medical care is an important component of evidence-based management of pediatric respiratory infections. Although there is a wide consensus that amoxicillin is the best antibiotic choice, there is marked variation in drug dosages and formulations between various guidelines [1,2].

Dosing of an antibiotic is an important determinant of its efficacy for the targeted bacteria at the site of infection. Insufficient dosing can lead to inadequate drug exposure against the bacteria and potential treatment failure. There have been relatively few well designed and conducted pharmacokinetic studies in children, to inform optimal weight-based dosing of antibiotics [3]. Over 70% of the available evidence of such dosing is from studies conducted with non-beta-lactam antibiotics [3]. Many oral formulations are still given as liquids, requiring the right reconstitution, correct storage after dilution, and right calibrated volume. Teaspoons are often used to judge the volume of reconstituted liquid medicine rather than calibrated bottle caps/droppers, causing medication errors [4]. The final dose administered to the child relies on effective communication between the prescriber and the carer. Finally, the acceptance of the child is dependent on the taste, consistency, and smell of the formulation. The WHO therefore recommends dispersible tablet formulation as the optimal method of providing medicines to children. The World Health Organization (WHO) has recently published guidance for primary care and hospital infections in children and adults, with a harmonized dosing schedule across infections [1]. In this paper, we focused on a single antibiotic, amoxicillin, and how the lack of dosing harmonization and convenient child-appropriate formulations in many countries challenges the standardization of treating children across the multiple clinical indications required for optimal drug procurement and program implementation.

## 2. Respiratory Infections in Children

Respiratory infections are common in children and most are caused by viruses. The most commonly implicated bacteria are *Streptococcus pneumoniae*, *Haemophilus influenzae*, *Moraxella catarrhalis*, and *Staphylococcus aureus* (Table 1) [5].

## 3. Amoxicillin

The Beecham Research Laboratories, USA, isolated the penicillin nucleus 6-aminopenicillanic acid (6-APA) in 1957. The isolation of 6-APA paved the way for the development of semi-synthetic penicillins, such as pheneticillin and methicillin in 1959 and ampicillin in 1961 [6]. Ampicillin’s limitations of poor oral absorption, shorter half-live, instability to beta-lactamase, and narrow spectrum were overcome by adding different side chains to 6-APA. Amoxicillin, (α-amino-p-hydroxybenzyl penicillin) developed in the early 1970s, was found to have a better oral absorption than ampicillin. Structurally similar to ampicillin, amoxicillin was also identified as having a similar spectrum of activity, which included *Enterococcus faecalis*, *Streptococcus pneumoniae*, beta-lactamase-negative strains of *Staphylococcus aureus*, *Streptococcus* spp., and beta-lactamase-negative strains of *Escherichia coli*, *Haemophilus influenzae*, *Neisseria gonorrhoeae,* and *Proteus mirabilis* [7]. Amoxicillin received FDA approval for use in children in 1974 and was marketed as “Amoxil” (capsule/oral suspension and chewable tablets), with a recommended daily dose of 20–40 mg/kg/day in three divided doses [8].

### 3.1. Clinical Pharmacology

The oral formulation of amoxicillin is absorbed immediately after consumption. Increasing the dose results in a longer time above the minimum inhibitory concentration (T > MIC), despite the non-proportional increase in maximum concentration [9]. It is excreted in urine. The hepatic metabolism of amoxicillin is poor, and about 60% of the oral dose is excreted unchanged through the renal system.

### 3.2. Mechanism of Resistance

Despite extensive aminopenicillin use in children for five decades, amoxicillin still retains its efficacy against most *Streptococcus* species, including *Streptococcus pneumoniae*. The mechanism of resistance is through modification of the penicillin-binding proteins, which reduces the affinity for beta-lactams. Table 2 provides the latest minimum inhibitory concentration (MIC) for amoxicillin from the European Committee on Antimicrobial Susceptibility Testing (EUCAST) guidelines and CLSI guidelines [10,11].

Unlike CLSI, EUCAST no longer uses an intermediate level of resistance for amoxicillin and simply classifies *Streptococcus pneumoniae* as susceptible or resistant [10]. In the CLSI guidelines (2022), the breakpoints for penicillin are defined according to the site of infection. For non-meningeal isolates, the breakpoints for amoxicillin are defined as susceptible ≤ 2 μg/mL, intermediate 4 μg/mL, and resistant ≥ 8 μg/mL. These breakpoints are applicable for an amoxicillin regimen of 500 mg administered every 8 h or 875 mg administered every 12 h [11]. There is limited country level data in children of *Streptococcus pneumoniae* resistance against penicillins. The median resistance rate of *Streptococcus pneumoniae* to penicillin reported by the GLASS surveillance in 2022 was low at 2.7% [IQR 0.3–13.8] (using non-meningeal breakpoints) [12].

### 3.3. Clinical Indications

Due to the continued clinical efficacy of amoxicillin against major respiratory pathogens, and its low toxicity and cost, amoxicillin is recommended as the first-line agent for treatment of acute respiratory infections by most national and international guidelines [1,2,13].

#### 3.3.1. Acute Otitis Media and Acute Bacterial Sinusitis

Acute otitis media and acute bacterial sinusitis usually occur as a secondary bacterial complication of a preceding viral upper respiratory infection. *Streptococcus pneumoniae*, *Haemophilus influenzae* (non-typeable), and *Moraxella catarrhalis* are implicated as common bacterial pathogens. The selection of resistant strains of *Streptococcus pneumoniae* in the middle ear and sinus cavities can occur due to prolonged nasopharyngeal carriage of *Streptococcus pneumoniae* and exposure to multiple courses of antibiotics. It was found that the penetration of penicillin into the middle ear fluid was lower compared to the lungs, which could potentially lead to treatment failure [14]. Amoxicillin doses at only 90–100 mg/kg/day were found to achieve middle ear fluid levels above the MICs of all intermediate and drug-resistant populations of *Streptococcus pneumoniae* [15]. Similarly, higher doses of amoxicillin were found to be required to eradicate drug-resistant *Streptococcus pneumoniae* from sinuses [16].

#### 3.3.2. Acute Pharyngitis/Pharyngotonsillitis

*Streptococcus pyogenes* is the leading bacterial cause of acute pharyngitis/pharyngotonsillitis in children. The minimum inhibitor concentration (MIC) breakpoints of amoxicillin for *Streptococcus pyogenes* (0.25) remain very low in comparison to *Streptococcus pneumoniae* (0.5). Even a single dose of amoxicillin at a dose of 40–50 mg/kg/day for 10 days was proven to be effective for eradicating Group A *streptococcus* (GAS) in pharyngitis/pharyngotonsillitis, instead of twice or three times daily dosing in children [17]. In countries with a high prevalence of rheumatic fever, a longer duration of 10 days, rather than 5 days for non-endemic countries, is used for the purpose of eradication of GAS.

#### 3.3.3. Community-Acquired Pneumonia

*Streptococcus pneumoniae* and *Haemophilus influenzae* are leading causes of community-acquired pneumonia. Production of beta-lactamases by *Haemophilus influenzae* confers resistance against Amoxicillin. However, a retrospective observational study showed that 80% of *Haemophilus influenzae* isolates, including those with severe pneumonia, were susceptible to amoxicillin [18]. The recent UK CAP-IT trial found that lower doses of amoxicillin, of 35 to 50 mg/kg per day instead of high dose of 70 to 90 mg/kg per day, and shorter duration (3 days instead of 7 days) were non-inferior in cases of non-severe pediatric community-acquired pneumonia in settings with a low prevalence of pneumococcal resistance [19]. Amoxicillin has been widely studied and was found to be effective in short durations in community-acquired pneumonia management in the low-middle income countries (Table 3) [20,21,22,23,24].

### 3.4. Standardization of Dosing Guidance for Multiple Infections

There is a wide range of doses used for amoxicillin globally. For example, for a 5 year old weighing 18 kg with non-severe pneumonia, dosing guidance ranges from 360 mg/day with a lower limit (20 mg/kg/day), as per Indian National guidelines, to 1620 mg/day using the higher limit (90 mg/kg/day) given in the ESPID Blue Book [13].

The Indian guidelines, for example, recommend low-dose amoxicillin (15–20 mg/kg/dose 12 hourly) for all respiratory infections [2]. The WHO has released guidance on the management of over 30 infections in children and adults, including the optimal antibiotic use, dose, and duration. To harmonize dosing across multiple infections, the AWaRe Book suggests simplified dosing of amoxicillin as 40–50mg/kg/dose 12 hourly for 5 days for all acute respiratory infections in children (see Table 4) [1].

### 3.5. Variation in Oral Formulations of Amoxicillin

#### 3.5.1. Available Strengths and Formulations of Amoxicillin

Amoxicillin is listed as an essential medicine by the WHO and is available in liquid form as 125 mg per 5 mL (as trihydrate) powder for oral liquid, 250 mg per 5 mL (as trihydrate) powder for oral liquid, and in solid form as 250 mg (as trihydrate) and 500 mg (as trihydrate) [25].

Amoxicillin is currently available in many countries in oral formulations as drops (100 mg/mL), i.e., as a bottle with dropper [26]; a suspension (125 mg/5 mL and 250 mg/5 mL); dispersible tablets (DT)(125 mg, 250 mg and 500 mg); and capsules (250 mg, 500 mg). Dispersible tablets are recommended by the WHO for all medicines for children, as they weigh less than bottles, do not require calculation by parents, and are simpler to store and distribute. In addition, patient compliance with dispersible tablets is easier to monitor, if needed [27]. Improved adherence to dispersible tablets compared to an oral suspension of amoxicillin has been reported, with an equivalent acceptability by young children and equivalent clinical outcomes for pneumonia [27]. The limitations are that the tablets require clean water for disintegration. The packaging of these tablets also needs to safeguard against the degradation of dispersible tablets under extreme temperatures and humidity conditions. To simplify the administration of amoxicillin in the field, innovative packaging has also been explored, including a 10 × 10 blister pack, a 1 × 10 blister pack, and a 2 × 10 blister pack of 250 mg DT amoxicillin [28]. However, in many countries dispersible tablets of amoxicillin are not widely available in the private sector. Of the over 400 brands of tablets/capsules available in a 250 mg strength, only about 25% of brands are available as DT, whereas only two of over 400 brands of 500 mg strength are available as DT [29].

#### 3.5.2. Cost

Costs vary significantly internationally. For example, in the Indian private market, DTs are slightly more expensive than suspensions; 250 mg DTs cost roughly INR 3–4 per tablet compared to INR 2/mL of 250 mg. However, through the UNICEF supply, the cost of Amoxicillin DT 250 mg is reduced to USD 0.27 for a pack of 1 × 10 (i.e., INR 2.2 per tablet) [28].

#### 3.5.3. Weight Bands for Dosing

Age-based dosing is significantly affected by the nutritional status of the child. Geographic variations in age by weight patterns could cause overdosing or insufficient dosing [30]. Therefore, the WHO recommends that weight-based bands are more appropriate for the identification of optimal doses, provided the current weight of the child is available. Since weight bands **(**Table 5) can provide a specific dose range in children, this approach clearly encourages the use of fixed-dose dispersible tablets [31].

#### 3.5.4. Administration of Higher Doses of Amoxicillin in ARI

If a child with an ARI weighs 10 kg, the dosing of amoxicillin using 50 mg/kg/dose would be 500 mg (i.e., 50 × 10) 12 hourly for 5 days. Compliance with such high dosages is difficult using liquid formulations but easier with 250 DT or 500 DT using weight-bands as in Table 5. Similarly, for a child weighing 20 kg, amoxicillin would be prescribed as 1000 mg/dose. Rather than giving four 250 mg DT of amoxicillin, a convenient dosing would be two 500 mg DT of amoxicillin 12 hourly. The ceiling dose of Amoxicillin is 2 g/day.

#### 3.5.5. Safety

Amoxicillin has a wide therapeutic index. Common adverse events include nausea and diarrhea, due to the disturbance of normal gut microbiota. Slightly higher rates of diarrhea may be associated with higher doses of amoxicillin (13.8% diarrhea; 6.5% rash) compared to low-dose amoxicillin (8.7% diarrhea; 2.9% rash). However, the diarrhea rate is lower than with other antibiotics, including the second-line antibiotic co-amoxiclav (18.9%) [32]. In the CAP-II trial, adverse events of diarrhea did not differ between those randomized to lower- versus higher-dose amoxicillin [19].

## 4. Summary

Amoxicillin is the first-line agent for the treatment of most common respiratory infections in children that require an antibiotic. However the dosing, dosing interval, and duration differ considerably between the various standard guidelines. There is evidence for variable dosing between different clinical indications. Harmonization of the dose across all respiratory infections could be achieved by using high-dose amoxicillin at 40–50 mg/kg/dose 12 hourly, as recommended by the WHO (Figure 1).

There is a complex balance between the simplicity of standardizing the dosing of medicines across different clinical infections and lower and higher dosing. The wide availability of simplified dosing regimens based on just two formulations, scored 250 mg and 500 mg dispersible tablets of amoxicillin, (Table 5) would ensure access to essential antibiotic treatment of all clinical respiratory tract infections, balancing effectiveness, toxicity, resistance, and cost-effectiveness.

## Figures and Tables

**Figure 1 antibiotics-12-01138-f001:**
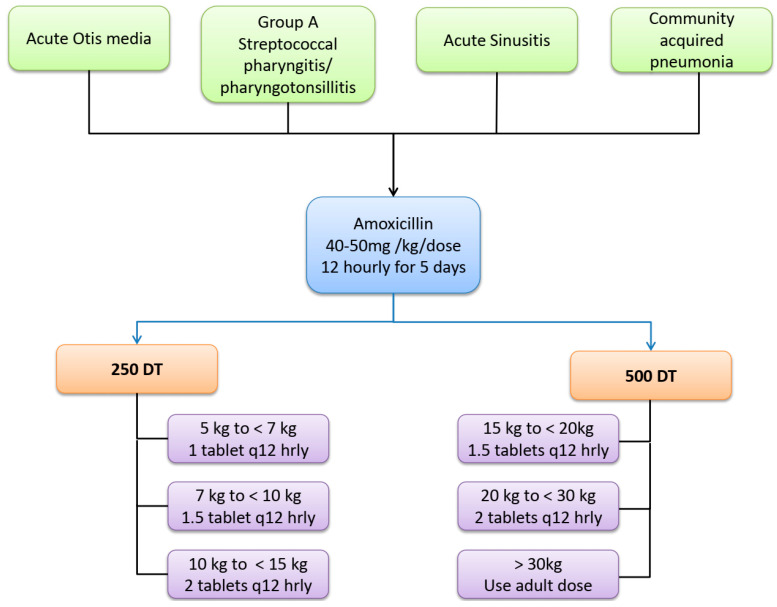
Suggested harmonization of amoxicillin dose, duration, and formulation for acute respiratory infections in children. (DT-Dispersible tablets). Longer duration of 10 days would be required for the treatment of Group A streptococcal pharyngitis in countries endemic for rheumatic fever.

**Table 1 antibiotics-12-01138-t001:** Respiratory infections in children with leading bacterial pathogens, in order of prevalence [5].

Infection	Leading Bacterial Respiratory Pathogens
Acute pharyngitis/pharyngotonsillitis	Group A β-hemolytic *streptocococci*
	Others:
	Group C and G streptococci, *Mycoplasma pneumoniae*,
	*Neisseria gonorrhoeae*, *Corynebacterium*
	*diphtheriae.*
Acute Otitis media	*Streptococcus pneumoniae*,
	*Moraxella catarrhalis*
	Non-typable *Haemophilus influenzae*
	Rarely
	*Streptococci* spp., *Staphylococci aureus*, *Pseudomonas*
Acute sinusitis	*Streptococcus pneumoniae*,
	*Moraxella catarrhalis*
	Non-typable *Haemophilus influenzae*
	*Streptococcus pyogenes*
Community Acquired pneumonia	**0–2 months**
	Gram-negative organisms
	Group B *Streptococcus*
	*Streptococcus pyogenes*
	Chlamydia
	**2 months–5 years**
	*Streptococcus pneumoniae*
	*Haemophilus influenzae*
	*Staphylococcus aureus*
	*Streptococcus pyogenes*
	*Mycoplasma pneumonia*
	**Above 5 years**
	*Streptococcus pneumoniae*
	*Staphylococcus aureus*
	*Streptococcus pyogenes*
	*Haemophilus influenzae*
	*Mycoplasma pneumoniae*
	*Chlamydia pneumoniae*

**Table 2 antibiotics-12-01138-t002:** Species specific current amoxicillin breakpoints for selected pathogens in EUCAST and CLSI guidelines, 2022 [10,11].

	EUCAST 2022		CLSI 2022		
Species	MIC	Interpretation	MIC	Interpretation	Comment
*Staphylococcus* spp.*	≤0.125 >0.125	S R	≤0.12 ≥0.25	S R	
*Streptococcus pneumoniae*	≤0.5 >1	S R	≤2 4 ≥8	S I R	Applicable for non meningial isolates for amoxicillin regimen of 500 mg administered every 8 h or 875 mg administered every 12 h
*Group A Streptococcus*	≤0.25 >0.25	S R	≤0.12 0.25–2 ≥4	S I R	
*Haemophilus influenzae*	≤0.001 >2	S R	≤1 2 ≥4	S I R	

* Most Staphylococcus species are penicillinase producers and some are methicillin resistant. MIC—minimum inhibitory concentration (in μg/mL); EUCAST—European Committee on Antimicrobial Susceptibility Testing; CLSI—Clinical and Laboratory Standards Institute; S—susceptible; I—intermediate; R—resistant.

**Table 3 antibiotics-12-01138-t003:** Randomized control trials on amoxicillin in community-acquired pneumonia from low-middle income settings, published between 2020 and 2022 [20,21,22,23,24].

Authors	Study	Age Group	Doses Given	Setting	Conclusions
[20]	Randomized control trial of amoxicillin versus placebo for management of pneumonia diagnosed by WHO criteria of tachypnea	2 to 59 months	According to WHO weight bands 40–50 mg/kg/dose q 12 hourly	Karachi	Difference in the treatment failure did not confirm the inferiority of placebo. A total of 44 children needed to be treated to prevent one treatment failure, indicating opportunities for improving antibiotic stewardship.
[21]	Cluster randomised trial for vommunity-based amoxicillin treatment for fast breathing pneumonia.	7–59 days	WHO weight bands: 40–50 mg/kg/dose q 12 hourly (125 mg two times per day for <4 kg body weight and 250 mg two times per day for >4 kg body weight.)	rural Bangladesh, Ethiopia, India and Malawi	A 7-day amoxicillin treatment for 7–59 days old non-hypoxaemic infants with fast breathing pneumonia by community level health workers was non-inferior to the currently recommended referral strategy.
[22]	Prospective, single blinded, parallel design, randomized controlled trial, efficacy of oral amoxicillin versus parenteral ceftriaxone in treatment of uncomplicated community-acquired pneumonia.	6 months to 12 years	100 mg/kg/day in three divided doses for 7 days.	tertiary centre, Mumbai, India	Use of oral amoxicillin for uncomplicated community-acquired pneumonia in children had a similar outcome as compared to parenteral ceftriaxone. 4% of children required step up to high dose amoxicillin.
[23]	Unblinded, cluster-randomized, controlled-equivalency trial for comparison of 3 days amoxicillin versus 5 days co-trimoxazole for treatment of fast-breathing pneumonia by community health workers.	2–59 months	50 mg/kg/day for 3 days	Haripur District, Pakistan	A 3-day course of oral amoxicillin was effective and safe treatment for fast-breathing pneumonia
[24]	Randomized control trial of Amoxicillin for 3 or 5 days for chest-indrawing pneumonia.	2–59 months	40 mg/kg/dose twice a day for 3 days versus 5 days	Outpatient departments in Malawi, Africa	Treatment with amoxicillin for chest-indrawing pneumonia for 3 days was noninferior to treatment for 5 days.

**Table 4 antibiotics-12-01138-t004:** Amoxicillin dose and duration recommended by the WHO, Indian Government, and IAP [1,2,5].

Clinical Conditions	WHO	Indian National Guidelines (ICMR) 2019	Indian Academy of Pediatrics 2014
Acute Otitis media	Amoxicillin 40–50 mg/kg/dose oral 12 hourly Duration 5 days	Amoxicillin 20 mg/kg/dose oral 12 hourly Or Co-amoxiclav 15–20 mg/kg/dose oral of amoxicilin component 12 hourly. Duration: 5–7 days In severe cases/children less than 2 years–10 days	Amoxicillin 20 mg/kg/dose oral 12 hourly Or Co-amoxiclav 45 mg/kg/day dose of amoxicllin component in two divided doses. Duration: 10 days
Group A streptococcal pharyngitis/tonsillitis	Amoxicillin 40–50 mg/kg/dose 12 hourly Duration 5 days (10 days in areas with high risk of Rheumatic fever)	Amoxicillin 15–20 mg/kg twice daily oral Duration 10 days	Amoxicillin Dose 50 mg/kg /dose 12 hourly Duration 10 days
Sinusitis	Amoxicillin 40–50 mg/kg/dose oral 12 hourly Second choice: Co-amoxiclav 40–50 mg/kg/dose (amoxicillin component) 12 hourly or 30 mg/kg/dose 8 hourly Duration 5 days	Amoxicillin 15–20 mg/kg twice daily Coamoxiclav 15–20 mg/kg/dose oral of amoxicilin component 12 hourly. Duration 10–14days	Amoxicillin 40 mg/kg/day in two divided doses If no improvement in 72 h or severe, Co-amoxiclav Duration 10 days
Mild to Moderate Community acquired pneumonia	Amoxicillin 40–50 mg/kg/dose 12 hourly Duration 5 days	Amoxicillin 15–20 mg/kg twice daily oral Co-amoxiclav 15–20 mg/kg of amoxicillin twice daily oral Duration 5 days	Amoxicillin 90 mg/kg/day (since penicillin resistant isolates are less than 10 percent, 50 mg/kg/day is sufficient) Or Coamoxclav Duration 5–7days

**Table 5 antibiotics-12-01138-t005:** Suggested dosing for amoxicillin using either 250 mg DT or 500 mg DT as first line for all respiratory infections in children (dosing around 40–50 mg/kg/dose 12 hourly) [1,31].

Weight Bands	Dose	Amoxicillin Formulation	Prescription
5 to <7 kg	250 mg q 12 hourly	250 DT	One tablet q 12 hourly
7 to <10 kg	375 mg q 12 hourly	250 DT	1.5 tablets q 12 hourly
10 to <15 kg	500 mg q 12 hourly	250 DT	2 tablets q 12 hourly
15 to <20 kg	750 mg q 12 hourly	500 DT	1.5 tablets q 12 hourly
20 to <30 kg	1000 mg q 12 hourly	500 DT	2 tablets q 12 hourly
30 kg and above	Use adult dose		

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
