# Peer review of "Harmonization of Amoxicillin Dose, Duration, and Formulation for Acute Childhood Respiratory Infections"

_antibiotics, 2023, doi:10.3390/antibiotics12071138_

Round 1
Reviewer 1 Report
Conclusion needs to be rewritten.
The manuscript format can be easily transformed into scoping review using Arksey and O Malley methodological framework.
Quality of English is acceptable but minor editing is recommended through professional editing services
Author Response
Thank you for the suggestions. We would like to retain as Viewpoint. As per your suggestion ,the manuscript has undergone grammar corrections through a professional software for language and grammar and further corrections made. The conclusion has also been modified with a flowchart of amoxicillin indications and recommended dosages.
Reviewer 2 Report
The view point entitled as “Harmonisation of amoxicillin dose, duration and formulation for childhood acute respiratory infections” by Dhanya Dharmapalan et al. Nevertheless, there are aspects that, in my opinion, should be worked on before publication.
I would shorten the abstract a little more and work out the essential points. Likewise, the question would be whether the title could be revised as “Harmonisation of amoxicillin dose, duration and formulation for acute childhood respiratory infections”.
.
The introduction section are clearly structured and provide a good introduction to the topic by including the indications of Amoxicillin in tonsillitis which is not mentioned please add. Also more recently published articles should be reviewed to improve the quality of the manuscript.
In the summary section, I would appreciate it if the flow chart could be made with dot representation for each individual indication of amoxicillin. This would provide greater clarity and transparency.
Author Response
Thank you for the suggestions. We have condensed the abstract to focus on the most important messages. The title has been revised as suggested. Indication for tonsillitis has been added. A flowchart for each indication of amoxicillin has been inserted in the summary.
Reviewer 3 Report
Manuscript ID: antibiotics-2421153, Title “Harmonisation of amoxicillin dose, duration and formulation for childhood acute respiratory infections”
The study of dose, duration and formulation optimization of amoxicillin is crucial as the safety and efficacy is concerned. However the therapy of amoxicillin must be based on the culture sensitivity test as well as the in vitro sensitivity pattern of amoxicillin through disk diffusion method. The culture sensitivity test must be carried out before start of therapy to know the resistance pattern of amoxicillin against various strains of bacteria. The dose, duration, dose frequency, and formulation dosage form may be selected on the basis of the resistance pattern.
The introduction section needs to be elaborated further. The materials and methods must be properly explained. The results must be represented in tables and graphs. The discussion section must be properly summarized.
The success of amoxicillin therapy must be evaluated on the basis of biomarkers and clinical investigations during the course of therapy. The duration and proposed intervention may be linked to these investigations.
Although the weight bands dispersible tablets are better alternatives instead of conventional formulations however the patient acceptability, ease of administration and patient compliance are the other parameters to be considered. The author must address these parameters in the manuscript. The outcomes of both the dosage forms (conventional vs DS) must be established on the basis of biomarkers, clinical investigations, patient acceptability and compliance.
The data of both the dosage must be compared and results may be drawn the basis of these studies. The compliance, efficacy, duration of therapy and cost-effectiveness of the therapies must be compared.
The author is required to incorporate some graphical data in the manuscript.
The manuscript must be revised as per journal approved format. The up to date references must be incorporated and grammatical and typographical mistakes must be rectified.
The language needs to be improved.
Author Response
We agree with the comments that an antibiotic needs to be selected on the basis of the resistance pattern of the pathogen and that the success of the therapy needs to be monitored clinically and with the help of biomarkers.
Amoxicllin is empirically recommended as a first line agent for acute uncomplicated respiratory infections by all current guidelines. It has undergone several randomised control trials for its efficacy (see table 3). Our paper aims to present a viewpoint on the need to harmonise the amoxicillin dosage which varies between guidelines and how best we can implement the recommendations using the formulation of dispersible tablets. As per the reviewer's suggestion, we have added the information on the comparison of dispersible and oral suspension concerning the acceptability and outcomes as follows: " Improved adherence to the dispersible tables compared to oral suspension of amoxicillin was reported with an equivalent acceptability by young children and equivalent clinical outcomes for pneumonia [27]" We have also added in the section 3.5.5 on safety comparison of the doses as " In the CAP-II trial, the adverse event of diarrhea did not differ between those randomized to lower- versus higher-dose amoxicillin[19]"
The language has been improved through a professional English software service and further corrections made.